# Efficacy of Supportive Care for Radiodermatitis in Patients with Head and Neck Cancer: Supplementary Analysis of an Exploratory Phase II Trial

**DOI:** 10.3390/jpm13091387

**Published:** 2023-09-16

**Authors:** Tsuyoshi Katsuta, Ikuno Nishibuchi, Megumi Nomura, Miho Kondo, Takao Hamamoto, Tsutomu Ueda, Bilegsaikhan Batsuuri, Takashi Sadatoki, Nobuki Imano, Junichi Hirokawa, Yuji Murakami

**Affiliations:** 1Department of Radiation Oncology, Graduate School of Biomedical Health Sciences, Hiroshima University Hospital, 1-2-3, Kasumi, Minami-ku, Hiroshima 734-8551, Japan; tkatsuta@hiroshima-u.ac.jp (T.K.); sadatoki@hiroshima-u.ac.jp (T.S.); imano@hiroshima-u.ac.jp (N.I.); junichih@hiroshima-u.ac.jp (J.H.); yujimura@hiroshima-u.ac.jp (Y.M.); 2Department of Nursing, Hiroshima University Hospital, 1-2-3, Kasumi, Minami-ku, Hiroshima 734-8551, Japan; migu@hiroshima-u.ac.jp (M.N.); konchan@hiroshima-u.ac.jp (M.K.); 3Department of Otorhinolaryngology, Head and Neck Surgery, Hiroshima University Hospital, 1-2-3, Kasumi, Minami-ku, Hiroshima 734-8551, Japan; takao0320@hiroshima-u.ac.jp (T.H.); uedatsu@hiroshima-u.ac.jp (T.U.); 4Department of Radiotherapy, National Cancer Center of Mongolia, Nam-Yan-Ju Street, Bayan Zurkh District, Ulaanbaatar 13370, Mongolia; beegii_baagii@yahoo.com

**Keywords:** head and neck cancer, radiotherapy, chemoradiotherapy, bio-radiation therapy, intensity-modulated radiation therapy, transcutaneous electrical sensory stimulation, acute radiation dermatitis, self-care, patient education, interferential current device

## Abstract

Self-care demonstrated efficacy in preventing severe acute radiation dermatitis among patients with head and neck squamous cell carcinoma undergoing chemoradiotherapy (CRT). This prospective trial aimed to confirm the feasibility and safety of transcutaneous electrical sensory stimulation while examining the relationship between changes in self-care behavior through supportive care interventions and the severity of acute radiation dermatitis during CRT. Patients underwent assessments for dermatitis grading (Grades 1 to ≥3) and were interviewed regarding self-care practices. The self-care questionnaires comprised six items, and a point was deducted for each task that the patient could not perform independently. Statistical analysis was performed to determine the association between G3 radiation dermatitis and the lowest self-care behavior scores. Of the 10 patients enrolled, three experienced G3 dermatitis. During CRT, six patients maintained their initial scores and did not develop ≥G3 dermatitis. Meanwhile, three of four patients with decreased scores exhibited ≥G3 dermatitis. The group with ≥G3 dermatitis had significantly lower scores than those with ≤G2 dermatitis, suggesting that the inability of patients to perform self-care routinely may lead to severe acute radiation dermatitis. Further prospective studies are needed to confirm the potential of self-care interventions in preventing severe dermatitis.

## 1. Introduction

Radiotherapy (RT) and chemoradiotherapy (CRT) are considered alternatives to surgery in patients with head and neck squamous cell carcinoma. Compared with radical surgery, RT/CRT is superior in preserving anatomical structures and organs. However, it has been known that irradiation of the head and neck region often develops adverse effects, such as acute mucositis, dermatitis, and pain. Acute radiation dermatitis is the most common adverse event, especially in cases treated with wide irradiation fields, such as bilateral multiple neck lymph node metastases [1]. Skin reactions include erythema, dry scaling, and itching in the early stages, followed by more severe reactions, such as wet scaling, wounds, and bleeding in later stages [2]. Since severe acute radiation dermatitis can occasionally cause systemic infection and treatment interruptions, dermatitis management during RT/CRT is crucial [3].

As the skin barrier function is impaired, the amount of moisture and sebum in the stratum corneum is reduced in acute radiation dermatitis [4,5]. Appropriate skin cleansing of the irradiated areas is highly recommended [6,7,8]. Washing the skin with warm water and foam hypoallergenic soap is recommended to avoid strong friction. Prophylactic topical application of moisturizers can prevent serious toxicity [6,9]. Patients usually perform preventive actions during CRT as part of their self-care. As self-care performance is largely self-dependent [9], patients should be responsible for maintaining a healthy state [10]. However, some patients may have difficulty in performing adequate self-care behaviors because of their age, activities of daily living, and cognitive function. It is important to assess the state of self-care and provide appropriate interventions in cases of deficiency. Adequate support throughout treatment can improve behavior and quality of life (QOL) [11,12]. 

In 2022, we launched a prospective study to evaluate the safety and feasibility of transcutaneous electrical sensory stimulation (TESS) therapy using an interferential current device in patients with head and neck squamous cell carcinoma who underwent CRT (TESS trial). The primary endpoint was the feasibility of TESS in head and neck squamous cell carcinoma patients undergoing CRT [13]. This was performed until the end of the treatment, and TESS did not have any adverse effects. 

Because the research participants were from the same population in which the radiation field was centered in a similar pharyngolaryngeal region, we also prepared issues for consideration. As there are few studies on the relationship between self-care and acute radiation dermatitis in the head and neck field, a secondary endpoint was established to evaluate the relationship between changes over time in self-care behaviors and levels through supportive care interventions and the severity of dermatitis in patients with head and neck squamous cell carcinoma.

## 2. Materials and Methods

### 2.1. Study Design Ethics

This prospective, single-center, single-arm study was initiated on 13 April 2022 and ended on 30 March 2023. This study was approved by the Certified Clinical Research Committee of Hiroshima University (certification number: CRB210005), registered with the Japan Registry of Clinical Trials (jRCTs062220008), and submitted to the Ministry of Health, Labor, and Welfare. This study was conducted in accordance with the tenets of the Declaration of Helsinki. Written informed consent was obtained from each participant before inclusion in the study.

This study was designed to evaluate the safety and feasibility of TESS using interferential current device in patients with head and neck squamous cell carcinoma undergoing CRT, and the details are described in an earlier report [13]. The enrolled patients underwent TESS to stimulate the muscles involved in swallowing movements to reduce dysphagia during CRT. The primary endpoint of this trial was to assess the feasibility rate. A supplementary endpoint was the achievement and efficacy of self-care behaviors with supportive care for patients with acute radiation dermatitis. 

### 2.2. Patient Selection

Patients were considered eligible if they had histological evidence of head and neck squamous cell carcinoma, and the tumor stage was classified according to the 8th edition of the American Joint Committee on Cancer Staging Manual and Handbook [14]. The eligibility criteria were as follows: (1) patients who underwent CRT for head and neck squamous cell carcinoma at Hiroshima University Hospital, (2) patients whose planned dose was 66–70 Gy to the laryngopharyngeal region, (3) patients aged >20 years at the time of consent, and (4) patients who provided written consent to participate in the study. Exclusion criteria were as follows: (1) patients with a history of RT in the head and neck area; (2) patients with tracheostomy; (3) patients in whom a part other than the laryngopharyngeal area had been irradiated; (4) patients with pacemakers and implantable cardioverter-defibrillators; (5) patients with difficulty placing interferential current device on the neck; (6) patients with many inconveniences in daily life (Eastern Cooperative Oncology Group (ECOG) performance status (PS) ≥2); (7) patients who were pregnant, may have become pregnant, or were breastfeeding; and (8) patients who were judged inappropriate by the principal investigator or the research coordinator.

### 2.3. Assessment, Intervention, and Data Collection

During CRT, patients were independently assessed by radiation oncologists and radiation oncology nurses every 10 Gy to determine the severity of acute neck radiation dermatitis. At the same time, the educational interview intervention consisted of approximately 20–30 min held face to face by the nurses. Self-care levels and behaviors were reviewed and assessed through interviews. The initial interview consisted of instructions on self-care behaviors, and at the end of the interview, a handout was given to the patient outlining the points to be aware of regarding care for dermatitis during treatment. Subsequent interviews focused on reviewing and confirming their behaviors.

Radiation dermatitis was recorded in accordance with the National Cancer Institute Common Terminology Criteria for Adverse Events (version 5.0). Grade (G) 1 was defined as the mildest disease and G5 as the most severe. Self-care behaviors for dermatitis consisted of instruction and confirmation of the following six items: (1) Patients understand that the irradiated area and their dermatitis are worsening. (2) Nail trimming. (3) Not wearing clothing with a collar that may irritate the skin. (4) The skin was washed with a soap layer to avoid rubbing it too hard. (5-a) Application of heparin analog ointment or dimethylisopropylazulene ointment without rubbing. (5-b) The fabric dressing was fixed (Moiskin pad, Hakujuji Corporation, Tokyo, Japan) with dimethylisopropylazulene ointment when the dermatitis became severe (Figure 1). (6) Being able to follow previous instructions regarding symptomatic treatments (5-a) and (5-b), only (5-a) was performed when the dermatitis was mild and (5-b) was performed when the side effect became severe; therefore, only one of these items was evaluated during the treatment period. For self-care behaviors, a score of 1 point was assigned to each item if the patient could fulfill it, and the total behavior score was recorded, with the highest score being 6 points and the lowest being 0 points. Scores were assigned every 10 Gy from 20 Gy to 70 Gy. 

A self-care judgment consisting of four levels (I–IV) was also performed. It was based on the Orem and Orem–Underwood self-care deficit nursing theories [15,16] and modified by Nakano et al. for practical clinical use [17]. The self-care judgements were as follows: Level I (wholly compensatory), Level II (a large portion compensated), Level III (a small portion compensated), and Level IV (independent).

### 2.4. Statistical Analysis

Quantitative data between self-care behavior scores and G3 were analyzed using the Mann–Whitney U test. Statistical significance was set at *p* < 0.05. Statistical analyses were performed using Excel statistical software package ver. 4.04 (Bellcurve for Excel; Social Survey Research Information Co., Ltd., Tokyo, Japan).

## 3. Results

Ten patients were enrolled in the study. The clinical data of the 10 patients are presented in Table 1. The median patient age was 67 (45–76) years, and the ECOG PS performance status score was 0. The primary sites were the nasopharynx (one patient), hypopharynx (seven patients), larynx (one patient), and unknown (one patient). Two, four, and four patients had clinical stages II, IVa, and IVb, respectively. Nine patients received a prescribed dose of 70 Gy/35 fractions, and one patient received 66 Gy/33 fractions. The patient was irradiated with a large field for an unknown primary cancer, resulting in severe mucositis, and RT was terminated based on clinical judgment. All patients were treated with intensity-modulated radiation therapy (IMRT) as the radiation technique. Combination chemotherapy consisting of cisplatin was administered to all patients except one who received cetuximab owing to decreased renal function.

Four patients were excluded from the study. There were two cases in which they refused to give their consent (they did not agree to continue hospitalization after CRT for rehabilitation purposes), and two cases in which RT was performed in areas other than the laryngopharyngeal region (both also included the region of the nasal cavity).

Among the ten enrolled participants, eight completed seven interviews from the 10 Gy point to the 70 Gy point. One patient (Patient J) skipped the first interview because of COVID-19. After recovery from the isolation period, an interview was conducted (20 Gy). In another patient (Patient C) who received 66 Gy, the last interview was not conducted at the patient’s request.

Acute radiation dermatitis, self-care levels, and self-care behavior scores evaluated every 10 Gy, starting at the 20 Gy time point, are listed in Table 2. Maximum dermatitis was observed in three patients in G1, four in G2, and three in G3. Regarding the background of the patients who became G3, Patient D received a cetuximab combination, developed delirium during treatment, and was treated by a liaison team. Patient G had dementia as a comorbidity. Patient H had a history of cerebral infarction and motor paralysis on one side from the beginning of treatment.

The mean score was 4.7 (range: 2–6) for the on-treatment self-care behavior score. Among all enrollees, six did not show a decrease in score and did not have acute radiation dermatitis above G3. Meanwhile, three of the four enrollees with reduced scores had dermatitis of ≥G3. The changes in self-care behavior scores for acute radiation dermatitis G3 are shown in Figure 2. The group with severe dermatitis had significantly lower scores than the group without severe side effects (mean score = 5.6 points in the <G3 group and 2.7 points in the ≥G3 group; *p* = 0.01) (Table 3).

Six assessments were conducted during CRT; the most difficult item to complete was item (5), application of ointment and fabric dressing. In contrast, item (2), trimming the fingernails, was the most satisfactory. The self-care level was maintained at Level IV in six patients until the end of treatment. One patient showed a decrease to Level III, and two showed a decrease to Level II.

## 4. Discussion

We conducted a phase II study on the efficacy of supportive care in patients with head and neck cancer undergoing CRT, with a 10% incidence of ≥G3 acute radiation dermatitis. Recent large CRT trials have reported that 20% of patients experience ≥G3 acute radiation dermatitis [18] and have suggested that supportive care may be effective in prevention. Six out of the ten patients did not show a decrease in the self-care behavior score and had experienced dermatitis grading <G3, while three out of four participants with reduced scores experienced severe dermatitis (≥G3). Our results show a correlation between self-care behavioral deficiency and severe dermatitis.

Our findings support the validity of the evaluation items used in the study by Jaarsma et al., which classified self-care behaviors into three categories, namely “self-care monitoring,” “self-care maintenance,” and “self-care management” [19]. Similarly, the six items used in this study can be classified under those three categories, with item 1 as self-care monitoring, items 2 to 5 as self-care maintenance, and item 6 as self-care management. The items were well balanced when observing the activities of the patient. The most common reason for the decline in the final score from the initial score was associated with failure to meet items 5-a and 5-b. These items pertained to the application of ointment and fixation of the gauze when dermatitis becomes severe, which is challenging to perform independently and correctly. Pictures and illustrations of correct ointment and gauze application can be applied in future studies to support patients to perform the task with ease. In contrast, item 2, which referred to trimming the fingernails, was the most fulfilled self-care activity in both initial and final score; however, this may be because the study was conducted with inpatients whose ward nurses regularly observed personal grooming. Therefore, the results may differ among outpatients. However, the validity of this study is likely to be the same because most CRTs for head and neck squamous cell carcinoma require an inpatient nurse for infusion. As a future prospect, a similar study for outpatients treated with RT alone should be conducted to confirm the effectiveness of supportive care. Compared to inpatient treatment, outpatient treatment limits the time available for nursing care. In recent years, the usefulness of telehealth interventions has been attracting attention [20], and the development of supportive care using various tools is expected in the future.

Radiation oncology nurses regularly assess developing dermatitis at the irradiated site and share preventive care and other symptoms to watch out for with patients. This information is valuable in determining the delay or progression of dermatitis. This review of evidence-based skin care plans reinforces patient education on strategies to prevent and manage side effects. Understanding skin care, prevention of skin damage, management of skin reactions, and activities that decrease symptoms empowers patients to engage in self-care and foster a sense of control throughout RT. Regarding postoperative RT for breast cancer, several studies have acknowledged the positive impact of patient education on skin care [12,21]. Although the management of dermatitis is an important issue in CRT for head and neck squamous cell carcinoma, there appears to be few reports on patient education in this area, which causes patient discomfort and reduces the QOL. Severe dermatitis can affect the effectiveness of RT by prolonging total treatment [22]. The favorable results of the present study showed the potential effectiveness of supportive care for the head and neck area. In the future, we would like to build on these results to conduct comparative studies and other trials with a higher level of evidence. In addition to head and neck squamous cell carcinoma and breast cancer, dermatitis is a major problem during RT for cervical esophageal cancer and anal canal cancer. Supportive care may be effective for these malignancies as well. Similarly, for pharyngitis and oral mucositis, various management actions have been advocated to prevent the worsening of symptoms [23]. It is also significant to confirm the need for patient education to ensure that these actions can be carried out correctly.

This study aimed to measure behavior scores and levels as part of a comprehensive assessment of the four-level classification of all self-care abilities, excluding specific behaviors. Given that many nurses work on rotating shifts, this classification serves as a useful criterion to rapidly share the self-care abilities of each patient and consider appropriate nursing interventions. According to the Orem–Underwood theory, only one level was available initially, described as “partially compensatory” [15,16]. However, many patients could be included in this stage in clinical practice. Therefore, Nakano et al. subdivided it into two levels, Levels II and III, according to the degree of compensation and developed judgment criteria [17]. In our study, no patient was classified as Level I, and loss of self-care ability was expressed as Level II or Level III. Therefore, it was appropriate to adopt the criteria of Nakano et al.

In this study, three patients had G3 acute radiation dermatitis. All patients had impaired motor or cognitive function due to pre-existing conditions or complications during treatment, and their self-care levels were reduced. We conducted interventions following the decline in their levels, considering the support they needed. However, there are limitations due to the limited time available. In the future, we aim to provide more effective supportive care for patients with declining levels. Patient E had no motor or cognitive problems but showed low scores and levels due to poor motivation for self-care. For this patient, a careful explanation of the significance of self-care behaviors and supportive motivation helped avoid severe side effects. Regarding populations of self-care with over 400 patients [24], they noted that patients who were older or less educated were less likely to practice self-care. In our study, since two of the three patients who had G3 acute radiation dermatitis were over 65 years of age, this report supports our findings. The authors also noted that less than half of all patients practiced self-care during a typical week of RT, suggesting that nursing intervention is necessary to ensure that care is practiced [24].

As a tool for functional assessment of the elderly, who are prone to lack of self-care, international cancer networks have recommended Comprehensive Geriatric Assessment as a key treatment strategy for all patients aged ≥70 years at the time of diagnosis [25]. It is defined as a multifaceted test battery that looks for signs of impairment across various age-related domains, such as comorbidity, function, physical performance, cognition, nutrition, emotional status, polypharmacy, social support, and living environment [26]. According to several trials, Comprehensive Geriatric Assessment is useful for identifying geriatric issues and predicting treatment tolerance, morbidity, and mortality in mixed cancer settings based on the patients’ functional ages. However, the time and resource demands involved in executing Comprehensive Geriatric Assessment have hampered its deployment in routine practice. Recently, however, it was found that the short Geriatric-8 screening instrument was reported to have its own the capacity to identify vulnerable patients and demonstrated predictive significance for functional decline and overall survival in a mixed oncology population [27]. The Geriatric-8 screening tool has been focused on its usefulness in the head and neck squamous cell carcinoma field. Patients with greater Instrumental Activities of Daily Living impairment were more likely to use gastrostomy tubes and less likely to regain QOL following treatment, according to research by VanderWalde et al. [28]. Neve et al. discovered that abnormal baseline Geriatric-8 scores were linked to lower completion rates of RT and worse postoperative outcomes [29]. Ishii et al. showed that the Geriatric-8 screening tool is a strong prognostic factor and predictor of complications in overall treatment, including RT [30]. For patients with low scores, enhancing support in advance might improve the maintenance of self-care behaviors. In addition, conducting this screening prior to RT could identify populations that need focused nursing intervention, which might lead to more efficient operations.

Radiation dermatitis toxicity has been reported to be influenced not only by the prescribed dose but also by advanced techniques, such as IMRT, to improve skin dose homogeneity [31] and radio-sensitizing systemic therapies (e.g., 5-fluorouracil, cetuximab) [32,33]. In this study, one patient received 66 Gy, all others received a uniformly prescribed dose of 70 Gy, and all patients received IMRT. Regarding systemic therapy, all but one of the remaining patients received concurrent cisplatin. Therefore, the characteristics of the participants in this study appeared to be homogeneous. One patient was treated with cetuximab as bio-radiation therapy (BRT) and developed G3. BRT is an effective option for patients who cannot receive CRT with platinum-based anticancer agents due to renal dysfunction or other reasons [34]. In contrast, adverse events often cause severe dermatitis; in the Bonner study, G3 and G4 were reported in 22.6% of patients [1] but later occurred more frequently than in previous reports [35,36]. In this case, the patient did not have self-care problems in the early stages of treatment, and skin reactions were mild at low doses; however, severe skin reactions occurred at higher doses. Therefore, more sensitive management of adverse events is required in cases of BRT.

This study had some limitations. First, the primary endpoint of this study was the feasibility and safety of TESS. This study was a sub-study that evaluated the severity of acute radiation dermatitis; however, TESS was performed with the potential to interfere with the skin. Second, this study was not a randomized controlled trial; therefore, we cannot determine whether supportive care actually reduces dermatitis. Third, the sample size was small. Fourth, the irradiation areas were not consistent. The areas of the neck irradiated with high doses differed due to the different N stages of each patient. Fifth, there is the possibility of strong patient selection bias due to a population of highly motivated patients who participated in the clinical trials and had good PS. However, the strength of this study was maintained as it was a prospective study. Dermatitis and patient interviews were conducted every 10 Gy to assess adverse events over time and confirm that supportive care was being administered. In addition, all patients were treated with IMRT, which was consistent with current standard irradiation techniques.

## 5. Conclusions

The association between a patient’s inability to perform self-care behaviors during treatment and the development of severe dermatitis has been documented. Therefore, multidisciplinary collaboration and encouragement become imperative when a decline in self-care behavior is observed. Future prospective studies must be conducted with a larger sample size and under more homogeneous conditions, focusing on the irradiated area. This will allow us to confirm whether severe acute radiation dermatitis can be avoided with supportive care intervention. Reductions in adverse events through supportive care can be applied to RT for malignant tumors other than head and neck squamous cell carcinoma and for adverse events other than dermatitis, which we would like to investigate in the future.

## Figures and Tables

**Figure 1 jpm-13-01387-f001:**
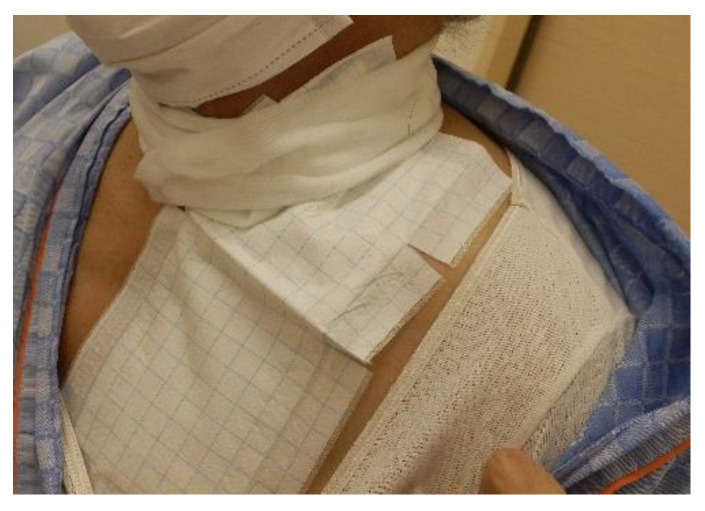
Fabric dressing fixation using dimethylisopropylazulene ointment.

**Figure 2 jpm-13-01387-f002:**
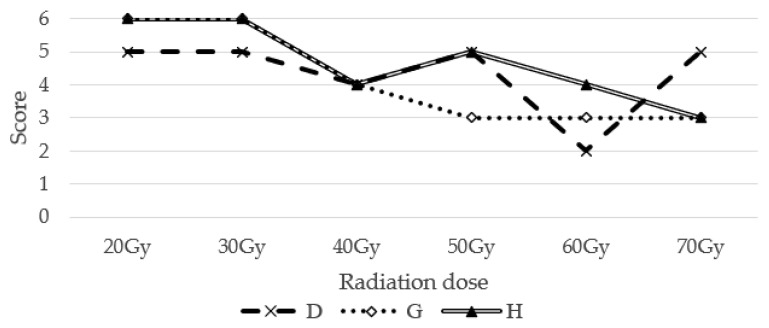
Change in self-care behavior score (acute radiation dermatitis ≥Grade 3).

**Table 1 jpm-13-01387-t001:** Characteristics of the patients enrolled.

Characteristic	No. of Patients (n = 10)	(%)
Sex		
Male	10	100
Female	0	0
Age, years		
Median	67	
Range	45–76	
ECOG PS		
0	10	100
1	0	0
Primary sites of cancer		
Nasopharyngeal	1	10
Hypopharyngeal	7	70
Laryngeal	1	10
Unknown primary	1	10
TNM stage		
II	2	20
IVA	4	40
IVB	4	40
Diabetes		
+	1	10
−	9	90
Radiation therapy		
70 Gy/35 fractions	9	90
66 Gy/33 fractions	1	10
Concurrent Chemotherapy		
Cisplatin	9	90
Cetuximab	1	10

ECOG PS, Eastern Cooperative Oncology Group performance status.

**Table 2 jpm-13-01387-t002:** Acute radiation dermatitis, self-care level, and self-care behavior scores for every 10 Gy.

J	I	H	G	F	E	D	C	B	A	Patient	
G1	G0	G1	G0	G0	G1	G0	G0	G1	G0	ARD *	20 Gy
IV	IV	III	IV	III	IV	III	IV	IV	IV	Self-care level	
6	6	6	6	3	6	5	6	6	6	SCBS	
G1	G1	G1	G0	G0	G1	G1	G1	G1	G0	ARD	30 Gy
IV	IV	III	IV	III	IV	III	IV	IV	IV	Self-care level	
6	6	6	6	3	6	5	6	6	6	SCBS	
G1	G1	G1	G1	G1	G1	G1	G1	G2	G0	ARD	40 Gy
IV	IV	III	III	III	IV	III	IV	IV	IV	Self-care level	
6	6	4	4	4	6	4	6	6	6	SCBS	
G1	G1	G2	G2	G1	G1	G1	G2	G2	G1	ARD	50 Gy
IV	IV	II	III	III	IV	III	IV	IV	IV	Self-care level	
6	6	5	3	4	6	5	6	6	6	SCBS	
G1	G1	G3	G3	G2	G1	G2	G2	G2	G1	ARD	60 Gy
IV	IV	II	III	III	IV	II	IV	IV	IV	Self-care level	
6	6	4	3	3	6	2	6	6	6	SCBS	
G1	G1	G3	G3	G2	G1	G3		G2	G2	ARD	70 Gy
IV	IV	II	III	IV	IV	II		IV	IV	Self-care level	
6	6	3	3	6	6	5		6	6	SCBS	

ARD, acute radiation dermatitis; * dermatitis was recorded according to the National Cancer Institute Common Terminology Criteria for Adverse Events (version 5.0); SCBS, self-care behavior score.

**Table 3 jpm-13-01387-t003:** Mann–Whitney U test of the association between self-care behavior score and Grade 3 or higher acute radiation dermatitis.

	ARD		
The Lowest Self-Care Behavior Score	<Grade 3	≥Grade 3	*p*-Value
Mean	5.6	3	0.01 *
Range	3–6	2–4	

ARD, acute radiation dermatitis; * *p* < 0.05.

## Data Availability

The data presented in this study are available upon reasonable request from the corresponding author.

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
