# Peer review of "Efficacy of Supportive Care for Radiodermatitis in Patients with Head and Neck Cancer: Supplementary Analysis of an Exploratory Phase II Trial"

_jpm, 2023, doi:10.3390/jpm13091387_

Round 1

Reviewer 1 Report (Previous Reviewer 2)

Although the manuscript refers to research that included a small number of respondents, the topic of the manuscript is interesting, the study is of a prospective nature and could be useful for some future research. Also, the authors have listed the shortcomings of the study at the end of the manuscript.

The authors fully responded to all the remarks of the reviewer. A modified version of the manuscript is fine.

Author Response

We wish to express our appreciation to Reviewer#1 for his or her insightful comments for the manuscript. In the future, as mentioned in the Discussion section, we would like to conduct a larger scale study.

Reviewer 2 Report (New Reviewer)

The authors acknowledge the weaknesses and limitations of this work. I absolutely share it.

I have proposed to the editors that they propose to the authors to transform the paper into a letter to the editor.

The abstract needs to be completely improved to make it more understandable.

Author Response

We wish to express our appreciation to Reviewer#2 for his or her helpful comments, which have helped us significantly improve the quality of our paper. Since you pointed out that the abstract was difficult to understand, we have revised it as follows for better clarity.
【Abstract】
(lines 18- 32) For patients with head and neck squamous…

Self-care demonstrated efficacy in preventing severe acute radiation dermatitis … to confirm the potential of self-care interventions in preventing severe dermatitis.

Reviewer 3 Report (New Reviewer)

Dear editor and dear authors,

Thank you for the opportunity to review your paper entitled "Efficacy of Supportive Care for Radiodermatitis in Patients with Head and Neck Cancer: Supplementary Analysis of an Exploratory Phase II Trial".

The introduction is well-framed. The authors explained the relevance of the present review and justified the theme.

The author could clarify the meaning of the last exclusion criteria in material and methods: "patients who are judged inappropriate by the principal investigator or the research coordinator."

The results and the discussion are well-framed. But in discussion, the most recent reference was from 2020, with a wide range between 1989 and 2020, which should be improved.

 The authors should develop the conclusion, explaining to the readers how these results can be applied in practice and what we should change after this research should be essential.

The manuscript has many acronyms, making it hard to read, so it is suggested to reduce them.

Author Response

Dear editor and dear authors,

Thank you for the opportunity to review your paper entitled "Efficacy of Supportive Care for Radiodermatitis in Patients with Head and Neck Cancer: Supplementary Analysis of an Exploratory Phase II Trial".

The introduction is well-framed. The authors explained the relevance of the present review and justified the theme.

The author could clarify the meaning of the last exclusion criteria in material and methods: "patients who are judged inappropriate by the principal investigator or the research coordinator."

  • Thank you for your supportive comments. We wish to express our appreciation to Reviewer#3 for his or her insightful comments, which have helped us significantly improve the quality of our paper.

    The results and the discussion are well-framed. But in discussion, the most recent reference was from 2020, with a wide range between 1989 and 2020, which should be improved.

    • Thank you for your pertinent question. As you have accurately pointed out, the range of ages of the papers used as references was wide. Therefore, we replaced the oldest papers with recent papers and added papers from 2022 as the most recent papers.
      【references】
      We replaced the papers as follows;
      Bataini, JP.; Asselain, B.; Jaulerry, C.; Brunin, F.; Bernier, J.; Pontvert, D.; Lave, C. A multivariate primary tumour control analysis in 465 patients treated by radical radiotherapy for cancer of the tonsillar region: clinical and treatment parameters as prognostic factors. Radiother Oncol. 1989, 14, 265-77. https://doi.org/10.1016/0167-8140(89)90138-2

      22. González Ferreira, JA.; Jaén Olasolo, J.; Azinovic, I.; Jeremic, B. Effect of radiotherapy delay in overall treatment time on local control and survival in head and neck cancer: Review of the literature. Rep Pract Oncol Radiother. 2015, 20, 328-39.  https://doi.org/ 10.1016/j.rpor.2015.05.010

      ※ In addition, the following references were added because a minimum of 30 references and a word count of 4000 characters were required.;
      【references】
      20. Kwok, C.; Degen, C.; Moradi, N.; Stacey, D. Nurse-led telehealth interventions for symptom management in patients with cancer receiving systemic or radiation therapy: a systematic review and meta-analysis. Support Care Cancer. 2022, 30, 7119-7132. https://doi.org/10.1007/s00520-022-07052-z. Epub 2022 Apr 14

      23 Owlia, F.; Kazemeini, SK.; Gholami, N. Prevention and management of mucositis in patients with cancer: a review article. Iran J Cancer Prev. 2012, 5, 216-20.
      24 Stake-Nilsson, K.; Gustafsson, S.; Tödt, K.; Fransson, P.; Efverman, A. A Study of Self-Care Practice in Routine Radiotherapy Care: Identifying Differences Between Practitioners and Non-Practitioners in Sociodemographic, Clinical, Functional, and Quality-of-Life-Related Characteristics. Integr Cancer Ther. 2022, 21, 15347354221130301.
      https://doi.org/10.1177/15347354221130301
      25. Wildiers, H.; Heeren, P.; Puts, M.; Topinkova, E.; Janssen-Heijnen, ML.; Extermann, M.; Falandry, C.; Artz, A.; Brain, E.; Colloca, G.; Flamaing, J.; Karnakis, T.; Kenis, C.; Audisio, RA.; Mohile, S.; Repetto, L.; Van Leeuwen, B.; Milisen, K.; Hurria, A. International society of geriatric oncology consensus on geriatric assessment in older patients with cancer. J Clin Oncol. 2014, 32, 2595-603.https://doi.org/10.1200/JCO.2013.54.8347
      26. Balducci, L. Supportive care in elderly cancer patients. Curr Opin Oncol. 2009, 21, 310–7. https://doi.org/10.1097/CCO.0b013e32832b4f25
      27. Hamaker, ME.; Jonker, JM.; de Rooij, SE.; Vos, AG.; Smorenburg, CH.; van Munster, BC. Frailty screening methods for predicting outcome of a comprehensive geriatric assessment in elderly patients with cancer: a systematic review. Lancet Oncol. 2012, 13, e437–44.
      28. VanderWalde, NA.; Deal, AM.; Comitz, E.; Stravers, L.; Muss, H.; Reeve, BB.; Basch, E.; Tepper, J.; Chera, B. Geriatric assessment as a predictor of tolerance, quality of life, and outcomes in older patients with head and neck cancers and lung cancers receiving radiation therapy. Int J Radiat Oncol Biol Phys. 2017, 98, 850–7. https://doi.org/10.1016/j.ijrobp.2016.11.048
      29. Neve, M.; Jameson, MB.; Govender, S.; Hartopeanu, C. Impact of geriatric assessment on the management of older adults with head and neck cancer: A pilot study. J Geriatr Oncol. 2016, 7, 457-462. https://doi.org/10.1016/j.jgo.2016.05.006
      30. Ishii, R.; Ogawa, T.; Ohkoshi, A.; Nakanome, A.; Takahashi, M.; Katori, Y. Use of the Geriatric-8 screening tool to predict prognosis and complications in older adults with head and neck cancer: A prospective, observational study. J Geriatr Oncol. 2021, 12, 1039-1043. https://doi.org/10.1016/j.jgo.2021.03.008

    The authors should develop the conclusion, explaining to the readers how these results can be applied in practice and what we should change after this research should be essential.

    You have raised an important question. The following information has been added to the Discussion and Conclusion from this result. (1) Supportive care may be indicated for other malignancies other than head and neck cancer with respect to dermatitis. (2) Supportive care may be effective for mucositis as well as dermatitis. We would like to continue our research on the above.
    【Discussion】
    (line 256) The following texts were added;
     As a future prospect, a similar study … is expected in the future.
    (line 276) The following texts were added;
     In addition to head and neck squamous cell carcinoma and breast cancer, … to ensure that these actions can be carried out correctly.
    【Conclusion】
    (line364~)The following texts were revised;
    Therefore, multidisciplinary collaboration ・・・ 

    Therefore, multidisciplinary collaboration and encouragement become imperative when a decline in self-care behavior is observed. Future prospective studies must be conducted with a larger sample size and under more homogeneous conditions, focusing on the irradiated area. This will allow us to confirm whether severe acute radiation dermatitis can be avoided with supportive care intervention. Reduction of adverse events by supportive care can be applied to RT for malignant tumors other than head and neck squamous cell carcinoma and for adverse events other than dermatitis, which we would like to investigate in the future.

    ※In addition, to increase the word count from the current manuscript, we have cited the literature on the relationship between the elderly and self-care and added a discussion of Comprehensive Geriatric Assessment.
    【Discussion】

    (line 302~) The following texts were added;
    Regarding the population in which …, which might lead to more efficient operations.

    The manuscript has many acronyms, making it hard to read, so it is suggested to reduce them..

    • We sincerely apologize for the difficulty in reading the manuscript due to the many acronyms. We have reduced the following abbreviations.
      【Abstract~Conclusion】
      SC⇒self-care
      ARD⇒acute radiation dermatitis
      HNSCC⇒head and neck squamous cell carcinoma
      IFCD⇒interferential current device

Round 2

Reviewer 2 Report (New Reviewer)

The paper has improved a lot.

Both the quality and the presentation of the results and the analysis of the conclusions.

I recommend its publication in the current format

This manuscript is a resubmission of an earlier submission. The following is a list of the peer review reports and author responses from that submission.

Round 1

Reviewer 1 Report

The authors present a paper about "Efficacy of Supportive Care for Radiodermatitis in Patients with Head and Neck Cancer: Supplementary Analysis of an Exploratory Phase II Trial".

The main idea is absolutely interesting and the topic totally deserves attention.

The introudction is well written and provides useful insights.

I have some concerns about the other sections of the paper as follows:

1) "patients who are judged inappropriate": what exclusion criteria would it be? it is absolutely not clear. How many patients were excluded due to this criteria? For which reason? This part needs addtional explanations

2) Were patients provided with a list of good practical rules to follow that they could adhere during the treatment? It is not adequately explained within the text this point, please provide additional information

3) I appreciate the honesty of the authors who declare that major limitations of this study include the high possibility of bias due to patient selection, the fact that irradiation areas were not consistent and most of all the extremely small size of patients included. It is absolutely difficult to say anythyng in favour or against the use of this protocol considering that only 3 patients experienced acute radiation dermatitis

4) I believe that no actual conclusions may be drwan on the few available data reported

Reviewer 2 Report

Congratulations to the authors on a brilliantly done manuscript!

The topic of the manuscript is current, and the manuscript itself is written very comprehensibly and clearly. I would especially like to praise the methodological part of the manuscript, which was done in detail.

Minimal corrections are required:

1. in the summary of the work, correct the following sentence: "The group of G3 had significantly lower scores than the group without (mean score."

2. delete the word "abbreviations" under each table, it is enough that they are listed

Reviewer 3 Report

This article was about the efficacy of supportive care in pts with head and neck cancer undergoing CRT. "It was a prospective trial that confirmed the role of transcutaneous electrical sensory stimulation" This sentence was reported in abstract, but TESS is not mentioned in any other way. 
The extent of skin toxicity of ENT tumours treated with CRT is well established.
This work is not an addition to the current knowledge.
Reading a paper on TESS is something I am looking forward to.
My hope is that the authors will be able to modify this work or resubmit it with greater attention and greater weight to innovative aspects. 

I recommend an accurate linguistic review
